

# Navigating challenges in radiography research: radiographers' perspectives in Saudi Arabia

Faisal A. Alrehily[1], Walaa Alsharif[1], Yasser Aloufi[1], Ammar Alsaedi[1], Marwan Aljohani[1], Reem S. Alotaibi[2], Hamed Alshammari[3], Abdullah Fahad A. Alshamrani[1], Fahad H. Alhazmi[1] and Abdulaziz Qurashi[1]

[1] Department of Diagnostic Radiography, College of Applied Medical Sciences, Taibah University, Madinah, Saudi Arabia
[2] Andalusia Hospital, Jeddah, Saudi Arabia
[3] Department of Radiological Sciences, Collage of Applied Medical Sciences, Imam Abdulrahman Bin Faisal University, Dammam, Saudi Arabia

Corresponding author
Faisal A. Alrehily,
frehily@taibahu.edu.sa

## ABSTRACT

**Introduction**. Radiography is a crucial healthcare specialty that requires ongoing research to advance imaging technologies and techniques. Despite this, radiographers are faced with obstacles such as time constraints, lack of resources, and the need for training on new technologies, which can discourage their research involvement. This study aims to provide a more representative understanding of the radiography research culture in Saudi Arabia, building upon previous studies.

**Methods**. Following the approval of an ethics committee at Taibah University (2024/173/302 DRD), a cross-sectional survey was conducted from January to March 2024 among registered radiographers in Saudi Arabia. An online questionnaire was distributed *via* social media platforms using a snowball sampling strategy to reach a diverse group of radiographers across different regions and institutions. Participants provided informed consent before completing the questionnaire. The questionnaire, originally in English, was translated into Arabic and validated by two bilingual academics. It included sections on demographics, previous research experience, barriers to research involvement, factors encouraging research engagement, and self-assessment of research competencies. Descriptive statistics and Cronbach's alpha were used to analyze the data.

**Results**. A total of 105 radiographers participated in the study, with 41% having prior research involvement. Among those engaged, the most common activity was data collection (65%), followed by preparation of scientific articles (49%). Challenges such as the lack of a research-focused culture (48%), insufficient awareness of opportunities (36%), and time constraints (34%) were prominent barriers to research participation. Encouraging factors included the need for research training (63%), support from research groups (51%), and allocated research time (50%). Respondents assessed their research skills, with confidence varied across skills, with 50% feeling capable of initiating research and 51% of participating, yet a significant proportion expressed uncertainties, especially in statistical knowledge and research methodology.

**Conclusion**. Most of the surveyed radiographers did not engage in research. However, there is a substantial interest in enhancing research involvement, with training, collaborative groups, and organizational support identified as key factors encouraging

participation. The findings suggest that addressing these barriers can foster a more robust research culture, leading to improved diagnostic practices.

## INTRODUCTION

Radiography is the cornerstone of modern healthcare. It plays a crucial role in disease detection, treatment guidance, and therapy effectiveness. This necessitates continuous and comprehensive research based on rapidly developing imaging technology. Such research is vital for advancing professional practice and providing evidence-based improvements in radiographic techniques (*Andersson, Lund En & Lundgren, 2020*; *Saukko et al., 2021*; *Törnroos et al., 2022*; *Vils Pedersen, 2023*). Generally, a robust research culture within health organizations may lead to greater service efficiencies, staff retention, and improved patient outcomes (*Harding et al., 2017*; *Scott, Waite & Napolitano, 2021*; *Neep, 2021*). These factors collectively underscore the importance of radiographic research, which is essential for improving diagnostic accuracy, and ensuring patients safety.

Although the value of radiographic research is widely acknowledged, radiographers often face various difficulties when attempting to conduct their own studies. These challenges can serve as significant barriers to engaging in research activities. Common obstacles include time limitations, a lack of resources, and the necessity to undergo formal training to work with new technology, such as artificial intelligence. These challenges may prevent their involvement (*Moran, Ab & Davis, 2020*; *Bolejko et al., 2021*; *Saukko et al., 2021*; *Chau et al., 2022*).

In Saudi Arabia, radiographers encounter similar barriers, along with cultural attitudes that marginalize their role, limited research opportunities, fear of failure, and a lack of collaborative environments (*Alshamrani et al., 2023*; *Alyami, Majrashi & Shubayr, 2023*; *Abuzaid et al., 2023*). Collectively, these factors impede radiographers' participation in research, underscoring the need for targeted interventions to address these challenges and foster a culture of research engagement within the profession.

Despite the recognition of these barriers, previous studies that investigated the involvement of radiographers in Saudi Arabia have been limited in their scope and generalizability. For instance, some researchers have focused on radiographers working across several hospitals of a single healthcare institution (*Alshamrani et al., 2023*). Conversely, while other studies have reported negative attitudes towards research among radiographers, they have not delved deeply into the underlying reasons for this reluctance across diverse work environments (*Alyami, Majrashi & Shubayr, 2023*), potentially failing to understand the reasons behind the low engagement among radiographers from a more diverse range of work environments. To address these gaps, the current study aims to offer a more comprehensive understanding of the radiography research culture in Saudi Arabia, expanding on previous research.

## MATERIALS & METHODS

A cross-sectional survey of registered radiographers in Saudi Arabia was conducted from January 2024 to March 2024. An online questionnaire was shared with radiographers through various social network platforms, including WhatsApp and Telegram. A snowball sampling strategy was used to encourage initial participants to share the questionnaire with their colleagues, which further extended its reach. To reduce potential bias, the distribution targeted a broad audience across different regions and institutions, inviting participation from radiographers with varying levels of experience. Before filling the questionnaire, the participants were asked to provide their consent to participate in the study. The study was approved by an ethics committee at Taibah University (2024/173/302 DRD).

The questionnaire was validated and used in previous studies (*Saukko et al., 2021*; *Chau et al., 2022*). The questionnaire was originally in English; however, to ensure the language barrier did not influence participation rate, the questionnaire was translated into Arabic. The translated questionnaire was reviewed and approved by two bilingual academics who are currently practicing in the radiography field and have over 10 years of experience.

The questionnaire was structured into several sections (Fig. 1). The first section included demographic information, such as age, gender, educational background, years of professional experience, and other relevant characteristics of the participants. The subsequent sections of the survey contained multiple-choice questions designed to assess the radiographers' previous research experience, documenting the nature of their prior involvement in research activities. This helped establish a baseline of research engagement within the group. Following that, the questionnaire addressed potential barriers to research involvement. It explored various factors that might discourage radiographers from participating in research, aiming to identify obstacles within the professional environment. The subsequent section was designed to pinpoint factors that could encourage radiographers to engage in research. The goal was to collect actionable insights that could help develop strategies to increase research participation. In these sections, respondents were asked to select the three most relevant answers from the provided options. Finally, the questionnaire concluded with items that allowed radiographers to evaluate their own research competencies using a 5-point Likert scale.

Descriptive statistics and frequency distributions were used to summarize the responses to the questionnaire. Cronbach's alpha was used to measure the internal consistency among the items included in the questionnaire.

## RESULTS

A total of 105 radiographers agreed to participate in the study. The majority of participants were male (58.10%) with 0–5 years of experience (64.76%). Detailed demographic information about the participants is presented in Table 1.

A total of 41% of the participants had been involved in research activities. Among these, data collection emerged as the most common task, with 65% of radiographers involved in this fundamental step. This was followed by the preparation of scientific articles and the development of research protocols, with 49% and 42% participation,

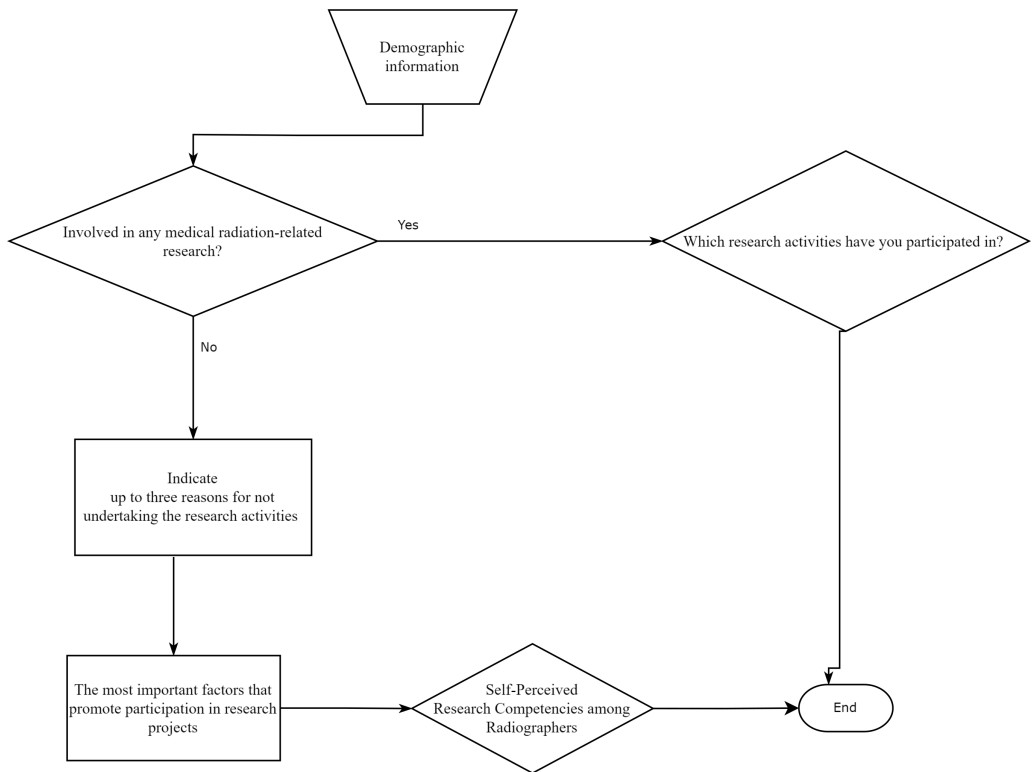

**Figure 1  Flowchart representation of the questionnaire used to explore research activity participation and barriers in the field of radiography.**

respectively. Data analysis also saw a participation rate of 42% among study participants. Ethical considerations, including applications to ethics committees, were undertaken by 35% of the respondents. Regarding the administrative aspects of research, 30% of radiographers participated in obtaining necessary hospital or other approvals essential for initiating research projects. However, activities such as presenting findings at conferences, reviewing background literature, and applying for project funding were less common, with involvement rates of 23%, 21%, and 21%, respectively (Fig. 2).

The study results indicated that 59% of participants had not previously engaged in research activities. The investigation into the barriers to research participation among radiographers revealed significant institutional and personal obstacles (Fig. 3). Notably, the lack of a research-focused culture at the workplace was the most cited obstacle, with 48% of participants identifying it as a major deterrent. Furthermore, 36% of respondents reported a lack of awareness of potential research opportunities, suggesting informational barriers within their professional environment. This issue is closely related to the reported lack of time, with 34% indicating that their schedules do not allow for engagement in research activities. Operational barriers were also noted, with a quarter of radiographers stating that research is not part of their designated work tasks. Challenges related to skills were mentioned by 18% of respondents, who feel inadequately equipped to participate in research. Similarly, a lack of personal interest in research was reported by another 18% of

**Table 1  Demographic information of the participants.**

| Characteristics | | Number of participants | The percentage (%) |
|---|---|---|---|
| Gender | Male | 61 | 58.10 |
| | Female | 44 | 41.90 |
| Age (years) | 18–24 | 21 | 20.00 |
| | 25–34 | 57 | 54.29 |
| | 35–44 | 25 | 23.81 |
| | 45–54 | 2 | 1.90 |
| Region | Eastern Region | 20 | 19.05 |
| | Middle Region | 13 | 12.38 |
| | Northern Region | 6 | 5.71 |
| | Southern Region | 25 | 23.81 |
| | Western Region | 41 | 39.05 |
| Employment status | Educational leave | 9 | 8.57 |
| | Full time | 96 | 91.43 |
| Highest qualification | Graduate diploma | 9 | 8.57 |
| | Bachelor's degree | 82 | 78.10 |
| | Master's degree | 8 | 7.62 |
| | Doctoral degree | 6 | 5.71 |
| Position | Radiographer in clinical practice | 99 | 94.29 |
| | Manager or equivalent | 6 | 5.71 |
| Workplace | Public hospital | 43 | 40.95 |
| | Semi-public hospital | 10 | 9.52 |
| | Private hospital | 52 | 49.52 |
| Years of experience (years) | 0–5 | 68 | 64.76 |
| | 6–10 | 13 | 12.38 |
| | 11–15 | 19 | 18.10 |
| | 16–20 | 5 | 4.76 |

participants. Additionally, 16% of participants expressed concerns about a lack of ideas for research projects, potentially indicating a need for more creative stimulation or ideation support. Minimal percentages (7% and below) reported a perceived lack of benefits from participating in research, a lack of enthusiasm within the community, difficulties obtaining ethical approval, or challenges in finding suitable research projects to participate in.

In terms of the factors that encourage involvement in research activities, 63% of respondents identified research training opportunities as a crucial resource. Membership in a research group was seen as beneficial by 51% of radiographers. Similarly, 50% of the participants indicated that having assigned time for research within their work schedules was essential. Support from colleagues outside of radiography was also notable, with 38% of respondents valuing this interdisciplinary backing. Organizational recognition was highlighted by 30%. Financial backing and material resource allocation were acknowledged by 30% of participants. Departmental managerial support was reported by 26% of respondents, while support from professionals in other institutions was cited by 25%.

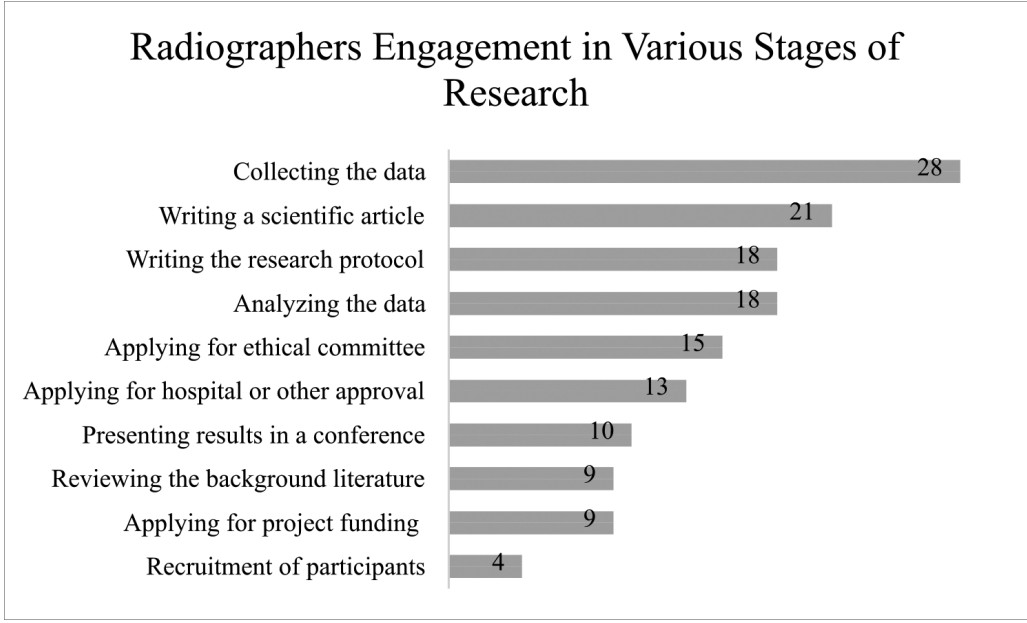

**Figure 2  The number of radiographers involved in different stages of the research process.** The number of radiographers engaged in various stages of research. The stages are listed from the highest to the lowest number of radiographers participating.

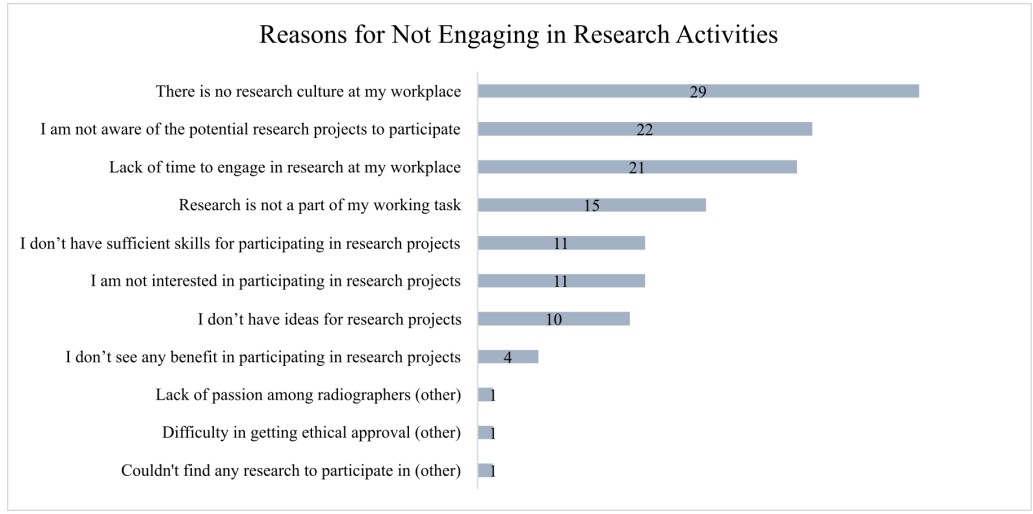

**Figure 3  Reasons for not engaging in research activities identified by radiographers who have not participated in research.** The reasons cited by radiographers for not engaging in research activities. The numbers indicate the number of radiographers who selected each reason.

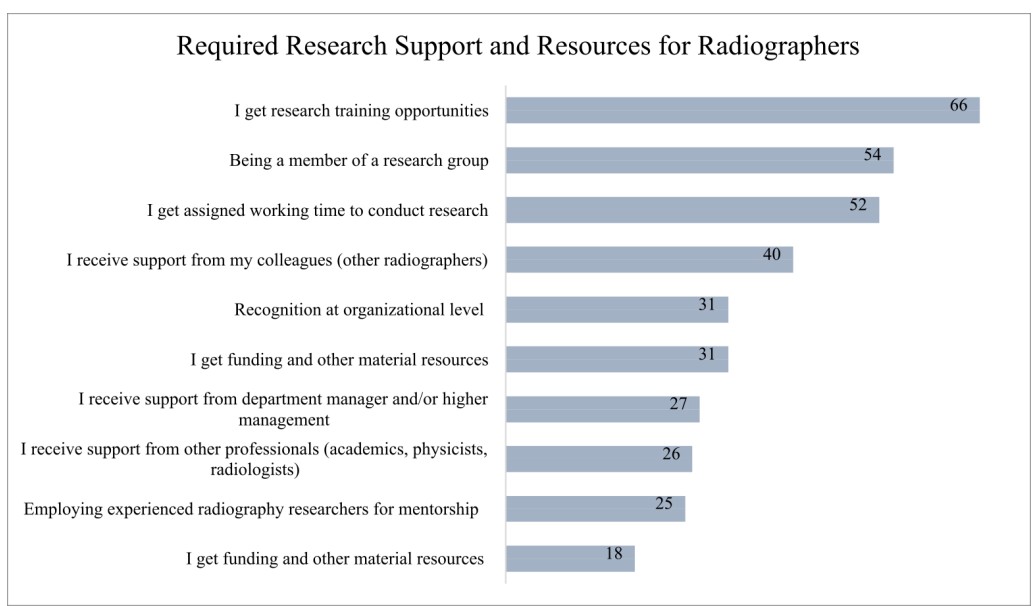

**Figure 4** **Key supports and resources identified by all the participants as essential for facilitating research engagement.** The types of support and resources required by radiographers to engage in research activities. The numbers next to each item represent the number of radiographers who indicated that particular type of support as necessary.

The recruitment of experienced radiographers for research positions was seen as beneficial by 24% of respondents. A smaller group, 17%, noted the requirement for additional funding and material resources (Fig. 4).

The self-assessed competencies in research-related skills among respondents were measured across eight items, and the value for Cronbach's alpha for the assessment was $\alpha = 0.95$. Figure 5 displays a gradient of confidence across various skill sets. A balanced proportion of respondents felt capable of both initiating (50% agree or strongly agree) and participating in (51% agree or strongly agree) research projects, with a smaller yet substantial fraction expressing reservation (21% neutral). Statistical knowledge appeared to be a notable area of uncertainty, with the highest percentage of respondents indicating neutrality (33%). Research methodology knowledge also showed the highest combined disagreement (29%). English language proficiency was considered adequate by a majority (54% agree or strongly agree). The ability to critically appraise research articles and conduct literature searches showed a moderate inclination towards agreement, yet a significant portion of respondents remained neutral (27% and 28%, respectively).

## DISCUSSION

There is a global necessity for healthcare professionals, including radiographers, to stay current with relevant research evidence to enhance their daily practice in radiology departments (*Ooi, Lee & Soh, 2012*; *Olive et al., 2022*). Conducting research in radiography is essential for the ongoing development and enhancement of diagnostic imaging, ultimately

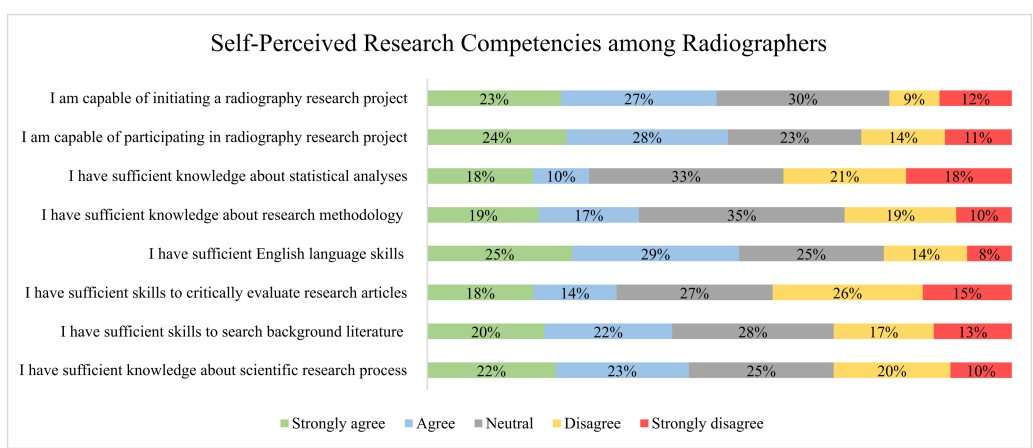

**Figure 5  Distribution of self-assessed research competencies among all particpants, highlighting areas of strength and potential development.**

benefiting patients and improving clinical outcomes. The study aims to understand the current status of research and the challenges faced by Saudi radiographers in initiating or enhancing their research activities.

The study results showed that 41% of radiographers in Saudi Arabia had engaged in research activities previously. Among those who participated, the primary form of engagement was in data collection. This focus aligns well with their clinical roles in hospitals, where they are often directly involved in the acquisition of diagnostic images and patient data. The limitation to such roles suggests that while radiographers are valuable contributors to the research process, their involvement is often not extended to other areas like data analysis, protocol development, or authorship of research papers (*Saukko et al., 2021*; *Alyami, Majrashi & Shubayr, 2023*; *Yakubu et al., 2023*).

However, when examining involvement in manuscript preparation, such as writing, study findings indicate a notable degree of involvement by radiographers in Saudi Arabia in the preparation of scientific manuscripts, with participation rates higher than those typically reported in the literature, which range from 19% to 36% (*Vikestad et al., 2017*; *Saukko et al., 2021*). This suggests a significant engagement in research activities compared to other regions, highlighting an increased emphasis on research within the radiography community in Saudi Arabia. However, our observation concerning the lack of detailed data on the specific roles played by radiographers in manuscript preparation is noteworthy. Without this information, it is difficult to fully assess the nature and impact of their contributions. Including more detailed questions in future surveys or research designs could provide deeper insights into the specific contributions of radiographers to scientific publications.

For those who have not participated in research activities, barriers cited include a lack of research-focused culture, heavy clinical workload, insufficient research resources, and inadequate funding. The study results are consistent with several studies from Saudi Arabia, Australia, Canada, the Nordic countries, and the United States (*Moran, Ab & Davis,*

*2020*; *Garlock-Heuer & Clark, 2020*; *Bolejko et al., 2021*; *Chau et al., 2022*; *Alshamrani et al., 2023*; *Alyami, Majrashi & Shubayr, 2023*; *Abuzaid et al., 2023*). This situation could be attributed to health organizations prioritizing clinical duties and patient care over research, accompanied by limited time allocation, managerial support, and resources for research activities, making it challenging to integrate research into regular practices (*Vils Pedersen, 2023*). Our study has also identified additional barriers to research participation, including a lack of awareness of research opportunities, insufficient skills, and difficulty generating ideas, which can be traced back to the prioritization of healthcare over research. Moreover, radiographers often encounter challenges with research skills due to a lack of confidence or the perception that research is irrelevant to their daily practice (*Saukko et al., 2021*). Communication barriers between academia and clinical practice could further impede the dissemination of information about research opportunities and ongoing projects. A study by *Williams, Craig & Robson (2020)* refers to the necessity for structured communication channels to bridge the gap between academia and clinical practice.

The study findings highlight critical areas for development in radiography research. Enhancing research training programs, promoting collaborative networks, fostering a supportive interdisciplinary environment, and, crucially, allocating dedicated time for research are pivotal strategies that can significantly improve the research culture among radiographers. These strategies not only support the professional development of individual radiographers, yet also advance the entire field, resulting in improved health outcomes and innovative practices (*Thingnes & Lewis, 2011*; *Hogg et al., 2020*). The study findings suggest that investing in these areas could yield substantial benefits in enhancing the research capabilities of radiographers.

This study is subject to several limitations that may impact the interpretation and generalizability of the findings. First, selection bias may have occurred due to the multi-platform distribution strategy employed to reach a diverse population of radiographers across different regions and institutions in Saudi Arabia. While this approach aimed to enhance diversity, it may have inadvertently excluded radiographers with limited access to technology or social media, potentially skewing the sample. Additionally, the reliance on self-reported data introduces the possibility of response bias, as participants might provide socially desirable answers or may not accurately recall their experiences. To mitigate this risk, we ensured the anonymity of the questionnaire and emphasized the confidentiality of responses; however, the potential for bias remains a concern. Furthermore, the adaptive nature of the questionnaire, where subsequent questions depended on prior responses, could have introduced measurement bias, despite our efforts to maintain consistency in question wording and format. We also acknowledge that while demographic information, such as age and gender, was collected, our analysis did not specifically examine their confounding effects on the results. Lastly, the study's small sample size, resulting from a low response rate, limits the generalizability of the findings to the broader population of radiographers. Future research should consider implementing strategies to improve response rates, such as follow-up reminders or incentives for participation, to obtain a more representative sample and enhance the reliability of the findings.

## CONCLUSIONS

The study offers a comprehensive overview of the current state of research involvement among radiographers in Saudi Arabia. The findings highlight a significant discrepancy between the potential for radiographers to contribute to research and their actual levels of engagement. The majority of radiographers have not previously participated in research activities, primarily due to substantial institutional and personal barriers. However, there is considerable interest in enhancing research involvement, as evidenced by the positive reception of research training, collaborative groups, and organizational support.

To cultivate a more robust research culture among radiographers, it is essential to address these barriers while leveraging the identified encouraging factors. By doing so, contributions of radiographers to research can be enhanced, ultimately leading to improved diagnostic practices and better patient outcomes.

## ACKNOWLEDGEMENTS

The authors extend their deepest gratitude to all the participants who contributed to this study. Their valuable inputs have been instrumental in enriching the authors' understanding of the research landscape within the radiography community. Additionally, this article was enhanced with the assistance of ChatGPT, which helped improve the clarity and coherence of the text.

### Funding

The authors received no funding for this work.

### Competing Interests

The authors declare there are no competing interests.

### Author Contributions

- Faisal A. Alrehily conceived and designed the experiments, performed the experiments, analyzed the data, prepared figures and/or tables, and approved the final draft.
- Walaa Alsharif conceived and designed the experiments, performed the experiments, authored or reviewed drafts of the article, and approved the final draft.
- Yasser Aloufi conceived and designed the experiments, performed the experiments, prepared figures and/or tables, and approved the final draft.
- Ammar Alsaedi conceived and designed the experiments, performed the experiments, prepared figures and/or tables, and approved the final draft.
- Marwan Aljohani conceived and designed the experiments, performed the experiments, prepared figures and/or tables, and approved the final draft.
- Reem S. Alotaibi conceived and designed the experiments, authored or reviewed drafts of the article, and approved the final draft.
- Hamed Alshammari conceived and designed the experiments, authored or reviewed drafts of the article, and approved the final draft.

- Abdullah Fahad A. Alshamrani conceived and designed the experiments, authored or reviewed drafts of the article, and approved the final draft.
- Fahad H. Alhazmi conceived and designed the experiments, authored or reviewed drafts of the article, and approved the final draft.
- Abdulaziz Qurashi conceived and designed the experiments, authored or reviewed drafts of the article, and approved the final draft.

## Human Ethics

The following information was supplied relating to ethical approvals (*i.e.*, approving body and any reference numbers):

Taibah University granted Ethical approval to carry out the study within its facilities (Ethical Application Ref: 2024/173/302 DRD).

## Data Availability

The raw data is available in the Supplemental File.

## Supplemental Information

Supplemental information for this article can be found online at http://dx.doi.org/10.7717/peerj.18125#supplemental-information.

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
