# Peer review of "Navigating challenges in radiography research: radiographers’ perspectives in Saudi Arabia"

_PeerJ, doi:10.7717/peerj.18125_

## Round 0.1 · original submission · Major Revisions

The authors are requested to carefully revise the manuscript and answer the questions raised by the reviewers.

Reviewer 1 ·

Basic reporting

Thank you for the opportunity for reviewing this paper.

The paper is well written.
The aim is interesting and important.
But there is some issues that need to be addressed.

Introduction
Consider including to reference some of the new papers from this year about this subject. I know Radiography have just published a paper investigating researchers’ perspective.
I think there is another Saudi study that included 5 Arab speaking countries – how does your study compare to this?
There are more recent studies from this year about AI and radiographers perception, consider to include them as references.
Please include the study aim

Method:
Please elaborate about the social network strategi. E.g. which platforms, and how did you make distribution (e.g. bias?).
How many questions in the survey?
Why not pilot the survey, if you translate the survey into Arabic?
How many items in the survey?
How many institutions were included – if relevant?
Informed consent?
Please explain the reply options/categories. Any dichotomy or was all multiple choice?

Results
Please include institutions or number of hospitals the respondents represented.
Any students?
Informed consent?
How does age and gender affect your results?
The results is only displayed with descriptive statistics. Consider including p-values.
Please correct table 1 – the first line/heading: -the percentag e (%)

Discussion
What does this study add to the already existing literature? What is new?
Especially since the title is “ a closer look” please include this.
Why repeat a study – please explain why this is relevant? And potential bias.

Please consider discussing your results with other Saudi studies – is the results similar (e.g. Abuzzaid et al).
Was this study distributed among students? The population is very young e.g. table 1.
Any differences between male and female’s radiographers? Please elaborate.
In the figures 1-4, please include number respondent and not just the %.
What field does the radiographers work in? Are they diagnostic radiographers or sonographers or?
Please include a section with your thought on bias in this study.

Experimental design

See comments

Validity of the findings

See comments

Reviewer 2 ·

Basic reporting

Somewhat clear and unambiguous and can be strengthened with regard to terminology use and lack of integration and synthesis. For example, throughout the manuscript it is unclear whether the target group were academics or clinically practicing radiographers. The title requires a review, that is, based on the questionnaire, which has items exploring the opportunities as well. Unsure about the phrase "closer look", appears to be a misfit.

The abstract -lines 23-46
Introduction: Could be strengthened with reordering the statements -lines 23-25. Then lines 25-27 already addresses the issues at hand, the why of the study become questionable. Lines 28-29 require a review after thorough read through of Alshamrani et al (2023) and Alyami, Majrashi & Shubayr (2023) findings.
Method: Is incomplete as per the standard requirement for an abstract
Results: Should focus on those with research involvement versus those who don't. Correlations would be helpful.
Conclusion: Weak with no unique scientific contribution in terms of evidence-based practice implications as well as clinical service delivery consequences. In other words, there should be pathway for practitioners who do not want to be engaged in research projects as such.

Manuscript
Introduction, lines 49-79
Can be strengthened with various types of research, dimensions of the research processes and associated complexities surrounding medical imaging. Provide a background on the profession in Saudi in terms of research undertakings and research development in progress both in the clinical and academic spaces. In other words, the achievements not only focusing on the challenges. Provide a literature overview nationally as well as globally on the progress made within the profession. Also, include arguments for and against in terms of the service delivery and resource implications as such.
Note, there is a very brief turnaround time between the two citations. Obviously, the results would be skewed as there is insufficient time for implementing the recommendations and suggestions made from the previous studies. Not sure and convinced about the contradiction and unsure of the concept "mixed", studies could be inconclusive, controversial or contradictory. Therefore, the aim for this study is weak need to be taught through.
There is a lack of logical flow of content thematically throughout this section.

Experimental design

Design meets the scope of the journal.
There is no research question.
Low to moderate regarding the research methods.

Materials and methods: lines 81-107 - There is no indication of the total of the targeted population group, for example, what is the national number of radiographers. Then is the target clinically practicing only, academic radiographers academic as well practicing or academics only. What is the sampling technique and size? There is no statistical evidence regarding this component. What are the eligibility criteria? The questionnaire was validated by two academics why not clinical practicing radiographers as well. What were the comments and amendments made, if any. How was the decision made to select only the two cited studies? Also, there is no information on the reliability of these selected study's instruments. The development of the instrument section needs some reworking in terms of the number of items and item selection of these cited studies.

Line 99 refers to systemic obstacles .... profession, not sure if this has been adequately addressed, need some rework to provide the context. Finally ..... line 102-103 ... there is an open-ended item which is not mentioned and not addressed in the result section as well. Descriptive statistics is insufficient because of the demographics. How did researcher ensure that participants who did take part in the previous studies were excluded from this study?

Conclusion: Insufficient

Validity of the findings

Results:
In the result section there is no mention made of the 59% participants with no prior research involvement and this comment also applies to Figure 1.
Would be great if the authors could draw a distinction between the non-research and the ones actively engaged then draw correlations.
Then also those who are undertaking postgraduate studies as such and or are academics.
If this was a baseline study the results would suffice, however, this a subsequent study as such and therefore below satisfactory.

Discussion: lines162-219- could be further condensed by avoiding duplication from the introduction section and also further integration and synthesis, especially where there are similarities with other studies. Not sure what is meant by lack of communication between academia and clinical practice, are post graduate qualifications not evidence thereof? Line 203 please check the citation.
Lines 206-212 highlights what is already known and well documented in literature.
Low participation could be accounted for the short space within which research studies on the same topic are being conducted. In actual fact is a confounding factor.
Conclusions: lines 223-232, the dilemma of quality-of-service delivery, the available resources and patient care considerations versus research. It is about striking that balance. So, in the introduction section of this manuscript that must be unpacked within the Saudi context as part of the rational/justification.
Conclusions: In the absence of a research question and or hypothesis, etc -weak

---

## Round 0.2 · Minor Revisions

The authors are requested to carefully revise the manuscript and answer the final issues raised by the reviewer.

Reviewer 1 ·

Basic reporting

ok

Experimental design

ok

Validity of the findings

ok. The findings could benefit of an gender analysis.

Additional comments

Peer J

Thank for this updated version of the paper.

The authors have addressed most of the questions.

Still the material and method section is very limited, and could with success have more data.

Reviewer 3 ·

Basic reporting

This is a revised submission and as such the work has already been assessed for many of the basic reporting areas. The revisions address the reviewers' comments in this area however the work would benefit from proofreading for English as there are several areas of awkward wording. For example, the committee providing approval for this study is referred to as an ethical committee rather than an ethics committee. The Methods section has some mixed tense usage, which is distracting.

In the Abstract it is unclear whether the results referring to challenges, barriers and encouraging factors apply to those doing research, those not engaged in research or both. This also needs to be made clear in the legends for Fig 3-5

Figures 2-4 – are the values on the plot number or percentage?

Experimental design

No comment - authors have addressed reviewer comments

Validity of the findings

No comment - authors have addressed reviewer comments

Additional comments

Overall the authors have addressed the reviewer comments with only some minor issues remaining

---

## Round 0.3 · Minor Revisions

The authors are requested to make final revisions to the figures as suggested by the reviewer.

Reviewer 3 ·

Basic reporting

No comment

Experimental design

No comment

Validity of the findings

No comment

Additional comments

The authors have addressed comments however I still feel that the Figures are unclear concerning whether data is number or percentage. For example, in Fig 4 the bars have numbers that could be percent or 'n' and it isn't clear which. This is further confused as Fig 5 shows the percentage. I suggest that the Figure legends be updated to include this information.

---

## Round 0.4 · accepted · Accept

After revisions, two reviewers agreed to publish the manuscript. I also reviewed the manuscript and found no obvious risks to publication. Therefore, I also approved the publication of this manuscript.

Reviewer 3 ·

Basic reporting

no comment

Experimental design

no comment

Validity of the findings

no comment

Additional comments

the authors have addressed all review comments